# Resistance Profiling and Molecular Characterization of Extended-Spectrum/Plasmid-Mediated AmpC β-Lactamase-Producing *Escherichia coli* Isolated from Healthy Broiler Chickens in South Korea

**DOI:** 10.3390/microorganisms8091434

**Published:** 2020-09-18

**Authors:** Hyun-Ju Song, Dong Chan Moon, Abraham Fikru Mechesso, Hee Young Kang, Mi Hyun Kim, Ji-Hyun Choi, Su-Jeong Kim, Soon-Seek Yoon, Suk-Kyung Lim

**Affiliations:** Bacterial Disease Division, Animal and Plant Quarantine Agency, 177 Hyeksin 8-ro, Gimcheon-si, Gyeongsangbuk-do 39660, Korea; shj0211@korea.kr (H.-J.S.); ansehdcks@korea.kr (D.C.M.); abrahamf@korea.kr (A.F.M.); kanghy7734@korea.kr (H.Y.K.); kimmh940301@naver.com (M.H.K.); wlgus01@korea.kr (J.-H.C.); kimsujeong27@gmail.com (S.-J.K.); yoonss24@korea.kr (S.-S.Y.)

**Keywords:** broiler, β-lactamase, *bla*_CTX-M_, *bla*_CMY-2_, *E. coli*, *mcr-1*, plasmids, quinolone

## Abstract

We aimed to identify and characterize extended-spectrum β-lactamase (ESBL)-and/or plasmid-mediated AmpC β-lactamase (pAmpC)-producing *Escherichia coli* isolated from healthy broiler chickens slaughtered for human consumption in Korea. A total of 332 *E. coli* isolates were identified from 339 cloacal swabs in 2019. More than 90% of the isolates were resistant to multiple antimicrobials. ESBL/pAmpC-production was noted in 14% (46/332) of the isolates. Six of the CTX-M-β-lactamase-producing isolates were found to co-harbor at least one plasmid-mediated quinolone resistance gene. We observed the co-existence of *bla*_CMY-2_ and *mcr-1* genes in the same isolate for the first time in Korea. Phylogenetic analysis demonstrated that the majority of *bla*_CMY-2_-carrying isolates belonged to subgroup D. Conjugation confirmed the transferability of *bla*_CTX-M_ and *bla*_CMY-2_ genes, as well as non-β-lactam resistance traits from 60.9% (28/46) of the ESBL/pAmpC-producing isolates to a recipient *E. coli* J53. The *ISECP*, *IS903*, and *orf477* elements were detected in the upstream or downstream regions. The *bla*_CTX-M_ and *bla*_CMY-2_ genes mainly belonged to the IncI1, IncHI2, and/or IncFII plasmids. Additionally, the majority of ESBL/pAmpC-producing isolates exhibited heterogeneous PFGE profiles. This study showed that healthy chickens act as reservoirs of ESBL/pAmpC-producing *E. coli* that can potentially be transmitted to humans.

## 1. Introduction

*Escherichia coli* is a commensal bacterium of the intestinal tract of humans and animals. It constitutes a reservoir of resistance genes for a wide range of pathogenic bacteria. The level of resistance in this bacterium is a good indicator of the selection pressure exerted by antimicrobial use and for the resistance problem to be expected in related pathogenic bacteria [1]. Therefore, investigation of the antimicrobial resistance profiles of indicator bacteria, such as *E. coli*, is essential to detect the spread of resistant bacteria between animals and humans [2].

Healthy food animals are frequently reported as reservoirs of extended-spectrum β-lactamase (ESBL) and plasmid-mediated AmpC β-lactamase (pAmpC)-producing *E. coli*, and have caught considerable attention worldwide [3,4]. The ESBL/pAmpC enzymes are known to hydrolyze the β-lactam ring of β-lactam antibiotics and cause the emergence of resistance to a considerable number of β-lactam antibiotics, including extended-spectrum cephalosporins [5]. Besides, ESBL/pAmpC-producing bacteria carry MDR genes, leaving only limited therapeutic options [6]. Human infections presumably occur following the ingestion of contaminated food of animal origin or via close contact with infected animals [7].

CTX-M-14 was the first CTX-M-type ESBL to be detected from isolates originated from food animals—i.e., *E. coli* isolated from chickens [8]. Since then, various types of CTX-M β-lactamases have been identified in *E. coli* recovered from food animals worldwide [9,10,11,12,13]. The distribution of CTX-M-type ESBLs varies depending upon geographical location. In the Republic of Korea (Korea), CTX-M-1, CTX-M-14, and CTX-M-15 are the most frequently detected ESBL types in isolates from food animals [14,15,16]. Recently, CTX-M-55 and CTX-M-65 ESBL types were noted in *E. coli* isolated from food animals and farm workers [16,17]. The observation implies that continuous surveillance of the phenotypic and molecular characteristics of ESBL/pAmpC-producing *E. coli* in food animals is vital to identify the prevalent ESBL/pAmpC phenotypes and to prevent the dissemination of β-lactam antibiotic resistance. Consequently, we undertook this study to provide new knowledge on the diversity of ESBL/pAmpC-producing *E. coli* isolated from healthy broiler chickens in Korea. Further investigations were also conducted to determine the mechanism(s) of the transfer of β-lactamases.

## 2. Materials and Methods

### 2.1. Collection of Samples and Isolation of E. coli

Fecal samples were collected from chickens originated from 34 broiler chicken farms located in six provinces of Korea in 2019. All broiler farms were conventional farms, with capacities of <50,000 (five farms), 50,000–100,000 (21 farms), 100,000–150,000 (six farms), and >150,000 (two farms) broilers. Cloacal swabs or fecal samples (8–12 samples per farm) were collected from six slaughterhouses using disposable sterile swabs. Samples were kept in an icebox and immediately transported to the Animal and Plant Quarantine Agency for further processing. The isolation and identification of *E. coli* were performed as described previously [18], using eosin methylene blue agar (EMB, Becton Dickinson, Sparks, MD, USA) and MacConkey agar plates (MAC, BD, Spark, MD, USA). Isolates were then confirmed by matrix-assisted laser desorption and ionization-time-of-flight mass spectrometry (MALDI-TOF, Biomerieux, Marcy L’Etoile, France). Only a single isolate per sample was considered for further assay.

### 2.2. Antimicrobial Susceptibility Testing

Antimicrobial resistance profiles of the isolates were determined by the broth microdilution method, according to the Clinical and Laboratory Standards Institute guideline (CLSI) [19], using commercially available Sensititre plates KRVP5F (Thermo Trek Diagnostics, Waltham, MA, USA). Sixteen antimicrobials were tested: amoxicillin/clavulanic acid, ampicillin, cefepime, cefoxitin, ceftazidime, ceftiofur, chloramphenicol, ciprofloxacin, colistin, gentamicin, meropenem, nalidixic acid, streptomycin, sulfisoxazole, tetracycline, and trimethoprim-/sulfamethoxazole. *E. coli* ATCC 25,922 and *E. coli* ATCC 35,218 were used as quality control strains. The interpretation of the results was according to the CLSI guidelines [19], the National Antimicrobial Resistance Monitoring System [20], and the European Committee on Antimicrobial Susceptibility Testing [21] guidelines. The MIC_50_ and MIC_90_ were calculated as the MIC that inhibited 50% and 90% of the isolates, respectively. Multi-drug resistance (MDR) was defined as resistance to at least three antimicrobial subclasses.

In addition, a double-disc synergy test was conducted to identify ESBL-producing isolates among ceftiofur-resistant *E. coli* using cefotaxime–cefotaxime/clavulanic acid and ceftazidime–ceftazidime/clavulanic acid discs (BD, Sparks, MD, USA), according to CLSI guidelines [19].

### 2.3. Detection of Resistance Genes

Polymerase chain reaction (PCR) assay was performed to detect the presence of *bla*_CTX-M_ genes using group-specific primers for CTX-M-1 and CTX-M-9. The complete *bla*_CTX-M_ was amplified and sequenced using previously-described primers. Additionally, a multiplex PCR assay was conducted to detect genes encoding for six AmpC families and positive isolates were amplified using specific primers. The *bla*_CTX-M_ and AmpC-positive strains were further screened for plasmid-mediated quinolone resistance (PMQR) genes: *qnrA*, *qnrB*, *qnrC*, *qnrD*, *qnrS1*, *qnrV*, *qepA*, and *aac (6′) Ib-cr* genes. Sequence analysis was performed using ABI3730XL DNA sequence analyzer (SolGent, Daejeon, Korea) and comparison with known sequences was performed with the Basic Local Alignment Search Tool (BLAST) programs at the National Center for Biotechnology Information website (www.ncbi.nlm.nih.gov/BLAST). The primers and their PCR conditions used for the detection of resistance genes are listed in Appendix A.

### 2.4. Conjugation Experiment

The broth-mating experiment was performed to determine the transferability of *bla*_CTX-M_ genes to sodium azide-resistant *E. coli* J53 [22]. Transconjugants were selected on Muller–Hinton agar, supplemented with sodium azide (150 μg/mL) and cefotaxime (2 μg/mL). The antimicrobial susceptibility profiles and β-lactamase gene carriage of the transconjugants were also determined, as described above.

### 2.5. Molecular Characterization of ESBL/pAmpC-Producing E. coli

A PCR-based replicon typing kit (DIATHEVA, Fano, Italy) was used to determine the replicon types of the transconjugants following the manufacturer’s protocol. The genetic environment of the *bla*_CTX-M/CMY-2_ genes was investigated using PCR and Sanger sequencing, as described previously [23,24]. A combination of *IS26* or *ISEcp1* forward primers, and a CTX-M reverse consensus primer (MA2) were used to investigate regions upstream of the *bla* genes. A *MA1* primer and reverse primers of *IS903* or *orf477* were used to characterize downstream regions of the *bla* genes. The primers and their PCR conditions used for the detection of the *bla*_CTX-M_ and *bla*_CMY-2_ genetic environments are listed in Appendix A. Additionally, pulsed-field gel electrophoresis (PFGE) analysis of ESBL/pAmpC-producing *E. coli* strains was also performed following *XbaI* digestion of chromosomal DNA (Takara Bio Inc., Shiga, Japan), as described previously [25]. Then, PFGE bands were analyzed using Bionumerics software (UPGMA) and relatedness of the isolates was calculated using the unweighted pair group method with the arithmetic average algorithm based on the Dice similarity index. Further, a multiplex PCR assay targeting *chuA*, *yjaA*, and the DNA fragment TspE4.C2 was used to determine the phylogenetic characteristics of the ESBL/pAmpC-producing strains [26].

## 3. Results

### 3.1. Antimicrobial Resistance of Indicator E. coli

We identified 332 *E*. *coli* isolates from 339 fecal samples obtained from 34 different broiler farms. Resistance to nalidixic acid (92.5%) was the highest, followed by resistance to ampicillin (86.4%), ciprofloxacin (78.3%), and tetracycline (71.7%) (Table 1). Resistance to amoxicillin/clavulanic acid, cefepime, cefoxitin, ceftazidime, and colistin was low (0.6–3.6%). We observed ceftiofur resistance in 13.9% (46/332) of the isolates. However, resistance to meropenem was not detected. All isolates were resistant to at least one antimicrobial agent, and MDR was noted in 94.3% of the isolates (Table 2). Besides, about 34% of the isolates exhibited resistance to at least eight antimicrobials. Among 103 different resistance patterns observed in this study, resistance to ampicillin, chloramphenicol, ciprofloxacin, nalidixic acid, streptomycin, sulfisoxazole, tetracycline, and trimethoprim/sulfamethoxazole (12.5%) was the most frequent MDR pattern.

### 3.2. Distribution of ESBL/pAmpC-Producing E. coli

We identified 46 (13.9%) ESBL/pAmpC-producing *E. coli* strains from 22 (64.7%) different broiler farms (Table 3). We observed four different types of ESBLs, namely CTX-M-55 (*n* = 18, 39.1%), CTX-M-14 (*n* = 12, 26.1%), CTX-M-1 (*n* = 4, 8.7%), and CTX-M-65 (*n* = 2, 4.3%). CMY-2 is the only pAmpC detected in eight (17.4%) *E. coli* strains, and two (4.3%) strains were positive for both CTX-M-55 and CMY-2 β-lactamases. The prevalence of ESBL/pAmpC-producing *E. coli* strains among farms ranged between 9.1% and 50%. Most farms carried one (59.1%) or two (36.4%) ESBL/pAmpC-types, while we noted three different ESBL/pAmpC-types from a farm in Jeonnam province.

### 3.3. Molecular Characteristics of ESBL/pAmpC-Producing E. coli

The ESBL/pAmpC-producing isolates exhibited resistance to several antimicrobial classes, such as aminoglycoside, tetracycline, quinolones, and folate pathway inhibitors (Table 4). Six of the CTX-M β lactamase-producing isolates were found to co-harbor at least one PMQR gene, with *qnrS*1, *qnrS2*, and *aac (6′)-Ib-cr* being detected alone or in combination. Notably, the *bla*_CTX__-M-65_, *qnrS2*, and *aac (6′)-Ib-cr* genes were found to be carried together in one isolate. Additionally, one isolate from farm E co-carried *bla*_CMY-2_ and *mcr-1* genes.

The *bla*_CTX-M_ and *bla*_CMY-2_ genes were transferred to recipient *E. coli* J53 from 55.3% (21/38) of *bla*_CTX-M_-positive (three *bla*_CTX-M-1_, five *bla*_CTX-M-14_, and 13 *bla*_CTX-M-55_) and 70% (7/10) of *bla*_CMY*-2*_-positive *E. coli* strains (Table 4). In addition, we observed the co-transfer of non-β-lactam antibiotic resistance, such as resistance to chloramphenicol, sulfisoxazole, tetracycline, and aminoglycosides along with *bla*_CTX-M_ and *bla*_CMY-2_ genes.

We identified various plasmid replicon types including IncI1α, IncFIB, IncFII, and IncHI2. IncFII (60.7%, 17/28) and IncI1α (35.7%, 10/28) were the most frequent plasmid replicon types (Table 4 and Table 5). Multiple replicon types were observed in 39.3% (11/28) of the transconjugants. Plasmids harboring *bla*_CTX-M-1_ and *bla*_CMY-2_ genes mainly belonged to IncI1α replicon type. Whereas, plasmids harboring *bla*_CTX-M-14_ and *bla*_CTX-M-55_ genes were predominantly associated with IncHI2 and IncFII replicon types, respectively.

The transconjugants carrying the *bla*_CTX-M-55_ gene presented distinct types of genetic environments, namely *bla*_CTX-M-55_-*orf477* (*n* = 9) and *ISEcp1*-*bla*_CTX-M-55_-*orf477* (*n* = 4) elements (Table 5). *ISEcp1*-*bla*_CTX-M-1_-*orf477* and *bla*_CTX-M-14_-*IS903* elements were identified in three and five transconjugants, respectively. The *bla*_CMY-2_ and *bla*_CTX-M-55+CMY-2_ gene expression was driven by the IS*Ecp1* insertion sequence, but *IS903* and *orf477* elements were not detected downstream of *bla*_CMY-2_ and *bla*_CTX-M-55+CMY-2_ genes.

PFGE analysis of 46 *E. coli* strains carrying *bla*_CTX-M_ and *bla*_CMY-2_ genes from 21 different farms demonstrated 34 arbitrary pulsotypes (Appendix A). In general, most of the isolates were heterogeneous. We observed identical PFGE profiles in *bla*_CTX-M-14,_
*bla*_CTX-M-55_, and *bla*_CTX-M-65_-carrying strains from farms AC, AE, AG, and I. Similarly, the two *bla*_CTX-M-14_-carrying isolates from farms AC and AE exhibited identical PFGE profiles. However, DNA from five strains was constantly auto-digested. Consequently, a cluster formed by these strains was excluded from the analysis.

Phylogenetic analysis of ESBL/pAmpC-producing strains showed that subgroup B1 was predominant (20/46, 43.5%), followed by A (20/46, 39.1%) and D (8/46, 17.4%). Notably, most *bla*_CMY-2_ carrying isolates (6/8, 75%) belonged to subgroup D.

## 4. Discussion

Our observations revealed that most of *E. coli* isolated from healthy broilers were resistant to multiple antimicrobials and possessed diverse ESBL-encoding genes that could be readily spread to humans. Although CTX-M-15 is considered the predominant ESBL type in the Korean poultry industry [27], we observed CTX-M-14 and CTX-M-55 type ESBLs in most of the isolates.

Consistent with previous findings in Korea [28,29] and other countries [30,31,32,33], *E. coli* isolates exhibited high rates of resistance to ampicillin, nalidixic acid, tetracycline, and sulfisoxazole. However, it was lower than those described in recent reports in Asia and Africa [34,35,36]. Additionally, the proportion of MDR isolates in this study corresponded with previous reports [28,33]. The isolates exhibited more than 100 different resistance patterns and most of these patterns were associated with quinolones, penicillins, and tetracyclines. High antimicrobial resistance rates and diverse resistance patterns observed in this study coincide with the marked increase in the use of antimicrobials, including penicillins, fluoroquinolones, phenicols, and tetracyclines in the Korean poultry industry [37]. The variations in antimicrobial resistance among countries might be because of differences in geographical region, locally approved antimicrobials, and farm management systems.

Fluoroquinolones are considered critically important antimicrobials for both humans and animals [38]. About 80% of the isolates were resistant to ciprofloxacin, a finding which is consistent with previous reports in Poland [32], Korea [39], and Vietnam [40]. However, it was higher than those reported in several Asian countries [31,35,41,42,43]. Although ciprofloxacin is not approved for animal uses, the continuous utilization of enrofloxacin in food animals, especially chickens in Korea, could be contributing to the increase in ciprofloxacin resistance [37].

Third-generation cephalosporin-resistant isolates are often resistant to multiple antimicrobials and are considered a potential threat to animal and human health [44]. The ceftiofur resistant rate in this study was slightly higher than previous reports in Korea (12%) [45] and the US (7%) [46]. Nevertheless, it was lower than Lee et al. (22%) [47] and Zhang et al. (47%) [33] in Korea and China, respectively. Various authors reported the relationship between ceftiofur use and resistance to third-generation cephalosporins in poultry production [48,49,50]. Therefore, although information on the use of this antimicrobial in farms was not available, the frequent application of ceftiofur in food animals could lead to the emergence of ceftiofur-resistant *E. coli* isolates.

A variety of ESBL/pAmpC genes have been identified in bacteria isolated from food animals worldwide. Most noteworthy of these are the *bla*_CTX-M-14_, *bla*_CTX-M-15_, *bla*_CTX-M-27_, and *bla*_CTX-M-55_ variants, which have been associated with the global spread of β-lactam antibiotic resistance in humans and food animals [51,52]. In Korea, β-lactam antibiotics resistance in chicken [14,16,28] and human [53,54,55] isolates is commonly associated with *bla*_CTX-M-1_, *bla*_CTX-M-14_, and *bla*_CTX-M-15._ However, *bla*_CTX-M-55_ was the most frequent ESBL gene observed in this study. Our finding concurred with a recent report in *E*. *coli* strains from retail chicken meat in Korea [17]. CTX-M-55 is a CTX-M-15 variant that possesses enhanced β-lactamase-hydrolyzing activity and structural stability [56]. Since its first detection in ESBL-producing *E. coli* in 2004 and 2005 in Thailand, it has been widely reported in *E. coli* isolated from food animals and humans in many countries [17,48,57,58,59,60]. The observation suggests that CTX-M-55 may be supplanting CTX-M-15.

*E. coli* harboring *bla*_CTX-M-14_ has been frequently detected in food animals in Korea [14,16,22] and other countries [57,60]. In this study, *bla*_CTX-M-14_ (26.1%) was the second most frequent ESBL gene. Similarly, Park et al. [17] and Seo et al. [28] detected *bla*_CTX-M-14_ in 22% and 14% of ESBL-producing broiler chicken *E. coli* isolates in Korea, respectively. Additional studies have also observed *bla*_CTX-M-14_-carrying *E. coli* isolates in food and companion animals, as well as in humans in several Asian countries [52,53,54,61,62], indicating its widespread distribution and the potential threat to public health.

In this study, only a few isolates were positive for *bla*_CTX-M-1_ and *bla*_CTX-M-65_. *bla*_CTX-M-65_ was frequently detected in ESBL-producing *E. coli* isolated from chicken in Korea [16,17] and China [52]. Although *bla*_CTX-M-1_ was detected in ESBL-producing strains recovered from chickens and farm environments in Korea [14,22,28], it is among the most frequent ESBL-encoding gene reported in Europe [61,63,64]. *blaCTX*_-M_ is known to spread between animals and humans through the food chain and isolates of humans and foods of animal origin commonly shared dominant CTX-M genotypes. Thus, broiler chickens may serve as an important reservoir and source of human infection [51].

pAmpC β-lactamase enzymes such as CMY-2 are less frequent in ceftiofur-resistant *Enterobacteriaceae* compared to the ESBLs [65]. We detected the *bla*_CMY-2_ gene in 21.3% of the ceftiofur-resistant isolates. Agreeing with this study, *bla*_CMY-2_ was the most common pAmpC β-lactamase-encoding gene in *E. coli* recovered from broiler chickens and humans worldwide [15,66,67,68,69,70,71]. The distribution of *bla*_CMY-2_ in several countries appears to be related to the efficient horizontal transmission of its encoding plasmids [72].

PMQR genes were commonly associated with low-level fluoroquinolone resistance and promoted the selection of high-level resistant strains [73]. In this study, the PMQR genes were identified in association with *bla*_CTX-M-1_, *bla*_CTX-M-55_, and *bla*_CTX-M-65_ genes. Most of the PMQR genes were associated with *bla*_CTX-M-55._ The *bla*_CTX-M-55_ genes commonly co-localize with other resistance genes, such as PMQR genes and genes encoding 16S rRNA methyltransferases [74,75]. The co-existence of PMQR and ESBL genes in *Enterobacteriaceae* have been reported in many countries, including Korea [73,76,77,78]. The widespread use of quinolones and third-generation cephalosporins in food animals has led to the emergence of PMQR and ESBL-producing *E. coli*. The co-occurrence of these genes in chicken isolates constitutes a public health concern.

The co-existence of ESBL and *mcr-1* genes in *Enterobacteriaceae* poses a serious public health threat. Despite several reports on the co-existence of *mcr-1* and ESBL genes in *E. coli* strains isolated from humans, food animals, and fresh vegetables in various countries [52,79,80,81,82,83], only a few reports are available on the co-existence of *mcr-1* and *bla*_CMY-2_ in *E. coli* [84,85,86]. Notably, this is the first report on the co-existence of *mcr-1* and *bla*_CMY-2_ in *E. coli* in Korea. Colistin is the last-resort antibiotic against multidrug-resistant *E. coli*, hence the co-existence of *mcr-1* and *bla*_CMY-2_ poses a serious challenge to the application of antimicrobials in humans and animals.

Various plasmid replicon types, either alone or in combination, were identified in *E. coli* transconjugants. Several studies have reported the association between *bla*_CTX-M-14_ gene and different plasmid types, including IncF family plasmids, IncK, and IncI1-Iγ [12,87,88]. However, this study identified the *bla*_CTX-M-14_ gene predominantly on the IncHI2 plasmid. The *bla*_CTX-M-55_ gene was efficiently transferred to recipient *E. coli* from 72% of *bla*_CTX-M-55_-carrying strains. This is presumably due to its frequent association with the IncF family of plasmids [27]. The IncF plasmid family is implicated in the dissemination of ESBLs because it is stably maintained in commensal *E. coli* [51]. In addition, the *bla*_CTX-M-1_ and *bla*_CMY-2_ genes predominantly belonged to IncI1α plasmid, a finding which concurred with Bevan et al. [51] and Carattoli, [7]. Further, the observation of diverse plasmid backbones in this study may reflect the co-occurrence of antimicrobial-resistant genes [27] and the dissemination of co-resistant bacteria [89].

ESBL-genes are often associated with insertion sequences (ISs), which are the smallest transposable elements capable of independent transposition in an organism [90]. The co-existence of *ISEcp1* and ESBL/pAmpC genes in *E. coli* isolates is well documented [90,91,92]. Agreeing with this study, IS*Ecp1* is frequently found in the upstream regions of ESBL/pAmpC genes and plays an important role in the efficient capture, expression, and mobilization of *bla*_CTX-M_ and *bla*_CMY-2_ genes [24,90]. Agreeing with previous reports [10,23,93], the *orf477* element was found downstream of *bla*_CTX-M-1_ and *bla*_CTX-M-55_ genes, while IS*903* was located downstream of *bla*_CTX-M-55_.

PFGE analysis demonstrated that the majority of the *bla*_CTX-M_-carrying isolates were highly diverse, except for specific clonal strains from the same or different farms, whereas all *bla*_CMY-2_-positive isolates showed different PFGE patterns. Therefore, clonal expansion and horizontal transmission within and between farms might contribute to the spread of ESBL/pAmpC-producing *E. coli* isolates. The proportion of subgroup D, which is considered pathogenic or an extraintestinal virulence-associated strain in our study (17.4%) was lower than Song et al. [27] (31%). The majority (82.6%) of ESBL/pAmpC-producing isolates in the current study mainly belonged to the commensal subgroups A or B1, which coincides with previous reports in Korea [27] and China [60]. Most of the pathogenic strains predominantly carried *bla*_CMY-2_, suggesting the emergence of pathogenic strains of *E. coli* carrying quinolone resistance genes in the Korean poultry industry.

In conclusion, our study showed that healthy broiler chickens were a major reservoir of *E. coli* that are resistant to multiple antimicrobials, including those ranked as medically important. This study identified ESBL/pAmpC-producing *E. coli* strains carrying predominantly *bla*_CTX-M-14_, *bla*_CTX-M-55_, and *bla*_CMY-2_ genes. Notably, the majority of *bla*_CMY-2_-carrying strains were pathogenic. This is the first report on the co-existence of *mcr-1* and *bla*_CMY-2_ in pathogenic *E. coli* in Korea. Both horizontal and clonal spread could be implicated in the dissemination of ESBL/pAmpC-producing *E. coli*. However, the multilocus sequence types of the isolates remained unclear. Altogether, the results suggest that healthy chickens are a matter of concern in terms of transmission of ESBL/pAmpC-producing *E. coli* to humans through the food chain. Therefore, the prudent use of antimicrobials in food animals is needed to prevent the introduction of ESBL/pAmpC-producing isolates into the food chain. Additionally, long-term surveillance is needed to trace the evolution and dissemination of ESBL/pAmpC-producing *E. coli* in food animals and its possible association with human isolates.

## Figures and Tables

**Table 1 microorganisms-08-01434-t001:** Antimicrobial resistance profiles and MIC distribution of *Escherichia coli* isolated from healthy broiler chickens in Korea.

Antimicrobials	Distribution (%) of MICs (µg/mL)													MIC_50_	MIC_90_	MIC	Resistance
	≤0.12	0.25	0.5	1	2	4	8	16	32	64	128	256	≥512			Range	(%)
Amoxicillin/clavulanic acid					3.6	10.8	69.9	12.3	3.3					8	16	2–32	3.3
Ampicillin					4.8	8.1	0.6			86.4				64	64	2–64	86.4
Cefepime		87.3	1.8	2.4	0.6	3	3.6	1.2						0.25	1	0.25–16	1.2
Cefoxitin					7.2	48.2	35.2	5.7	3.6					1	4	1–32	3.6
Ceftazidime				89.5	1.5	2.4	3	3.6						1	2	1–32	3.6
Ceftiofur			75	10.5		0.3	14.2							0.5	8	0.5–8	13.9
Chloramphenicol					0.3	12.7	18.4	1.5	2.1	65.1				64	64	2–64	67.2
Ciprofloxacin	1.8	11.7	4.8	1.5	1.8	2.7	43.7	31.9						8	16	0.12–16	78.3
Colistin					99.4		0.6							2	2	2–8	0.6
Gentamicin				70.2	3.9		0.3	2.4	3	20.2				1	64	1–64	25.6
Meropenem		100												0.25	0.25	0.25–0.25	0
Nalidixic Acid					2.1	1.8	2.1	1.5	0.6	1.5	90.4			128	128	2–128	92.5
Streptomycin								38.3	5.4	7.5	48.8			64	128	16–128	61.7
Sulfisoxazole								29.8	2.1				68.1	512	512	16–512	68.1
Tetracycline					28.3				14.8	22.6	34.3			64	128	2–128	71.7
Trimethoprim/	33.4	9.3	4.8	2.4	0.3	49.7								1	4	0.12–4	49.7
sulfamethoxazole																	

The dilution ranges tested are those contained in the white area. The breakpoints of tested antimicrobial agents are indicated by vertical lines. MIC_50_ and MIC_90_ are the concentrations at which 50% and 90% of the isolates were inhibited, respectively.

**Table 2 microorganisms-08-01434-t002:** Frequent antimicrobial resistance patterns of *Escherichia coli* isolated from healthy broiler chickens in Korea.

No. of Antimicrobials	Total No. of Isolates (%)	Frequent Resistance Pattern (No. of Isolates)
0	0 (0)	
1	1 (0.3)	NAL (*n* = 1)
2	18 (5.4)	AMP TET (*n* = 8)
3	30 (9)	AMP CIP NAL (*n* = 8)
4	26 (7.8)	AMP CIP NAL TET (*n* = 9)
5	53 (16)	AMP CHL CIP NAL TET (*n* = 16)
6	34 (10.2)	AMP CIP GEN NAL STR TET (*n* = 4)
7	58 (17.5)	AMP CHL CIP NAL STR FIS TET (*n* = 13)
8	55 (16.6)	AMP CHL CIP NAL STR FIS TET SXT (*n* = 39)
9	45 (13.6)	AMP CHL CIP GEN NAL STR FIS TET SXT (*n* = 35)
10	6 (1.8)	AMP XNL CHL CIP GEN NAL STR FIS TET SXT (*n* = 3)
11	2 (0.6)	AMC AMP FOX CAZ XNL CHL CIP NAL STR FIS TET (*n* = 1)
12	1 (0.3)	AMC AMP FOX CAZ XNL CHL CIP NAL STR FIS TET SXT (*n* = 1)
13	3 (0.9)	AMC AMP FOX CAZ XNL CHL CIP COL NAL STR FIS TET SXT (*n* = 1)
MDR (≥3 subclass)	313 (94.3)	

AMC, Amoxicillin/clavulanic acid; AMP, ampicillin; FOX, cefoxitin; CAZ, ceftazidime; XNL, ceftiofur; CHL, chloramphenicol; CIP, ciprofloxacin; COL, colistin; GEN, gentamicin; NAL, nalidixic acid; STR, streptomycin, FIS, sulfisoxazole; TET, tetracycline; SXT, Trimethoprim/Sulfamethoxazole. MDR; Multi drug resistant (resistance to 3 or more antimicrobial subclasses).

**Table 3 microorganisms-08-01434-t003:** Distribution of *bla*_CTX-M_ and *bla*_CMY-2_-carrying *Escherichia coli* isolates in broiler chicken farms in Korea.

Farm ID	Province	Prevalence (%) (No. of *bla-*Positive Isolates/ Total No. of Isolates per Farm)	CTX-M-1	CTX-M-14	CTX-M-55	CTX-M-65	CMY-2	CTX-M-55+CMY-2
2019-P-A	Chungbuk	12.5 (1/8)			1			
2019-P-C	Gyeongbuk	25 (2/8)	1		1			
2019-P-E	Gyeongbuk	27.3 (3/11)		1			2	
2019-P-G	Jeonbuk	10 (1/10)					1	
2019-P-I	Jeonbuk	50 (5/10)		4	1			
2019-P-K	Jeonbuk	20 (2/10)					2	
2019-P-M	Jeonnam	10 (1/10)			1			
2019-P-N	Chungbuk	10 (1/10)						1
2019-P-O	Gyeongbuk	10 (1/10)	1					
2019-P-P	Gyeongbuk	20 (2/10)		1	1			
2019-P-T	Chungnam	9.1 (1/11)	1					
2019-P-V	Jeonnam	10 (1/10)			1			
2019-P-W	Jeonbuk	30 (3/10)			2			1
2019-P-Y	Jeonbuk	10 (1/10)			1			
2019-P-Z	Chungnam	20 (2/10)	1				1	
2019-P-AA	Chungnam	9.1 (1/11)			1			
2019-P-AB	Jeonnam	10 (1/10)					1	
2019-P-AC	Jeonbuk	40 (4/10)		1	3			
2019-P-AD	Gyeonggi	30 (3/10)		2	1			
2019-P-AE	Jeonnam	40 (4/10)		1		2	1	
2019-P-AF	Jeonbuk	20 (2/10)		2				
2019-P-AG	Jeonbuk	40 (4/10)			4			
Total		22.4 (46/219)	4	12	18	2	8	2

The numbers in columns 4–9 represent the number of isolates that carried bla_CTX-M_ and/or bla_CMY-2_ genes.

**Table 4 microorganisms-08-01434-t004:** Characteristics of ESBL/pAmpC-producing *Escherichia coli* isolated from healthy broiler chickens in Korea.

Isolates	Farm ID	Provinces	MICs (µg/mL)			*bla* Gene	PMQR Gene	Non-β Lactam Antibiotic Resistance	Transfer- Ability	Plasmid Type	Phylogenetic Group	Pulsotype
			XNL	FOX	CAZ							
A-CF-6	A	Chungbuk	>8	4	4	CTX-M-55	-	CIP NAL	+	FⅡ	A	P22
C-CF-6	C	Gyeongbuk	>8	8	2	CTX-M-1	*qnrS1*	CHL GEN STR FIS TET SXT	+	I1α, FIB, FⅡ	A	P9
C-CF-8	C	Gyeongbuk	>8	8	4	CTX-M-55	*qnrS1*	CHL FIS	+	I1α, FⅡ	D	P6
E-CF-2	E	Gyeongbuk	>8	32	16	CMY-2	-	AMC STR FIS TET SXT	-	-	D	P31
E-CF-M-5	E	Gyeongbuk	>8	>32	16	CMY-2	*mcr-1*	CHL CIP COL NAL STR FIS TET SXT	+	I1α, FⅡ	D	P32
E-CF-8	E	Gyeongbuk	8	8	≤1	CTX-M-14	-	CHL CIP NAL FIS	+	HI2, Iγ, FⅡ	A	P26
G-CF-M-6	G	Jeonbuk	>8	>32	>16	CMY-2	-	CHL CIP NAL STR FIS TET SXT	+	I1α	A	P34
I-CF-1	I	Jeonbuk	>8	2	4	CTX-M-55	*qnrS1*	TET	+	FⅡ	B1	P7
I-CF-M-2	I	Jeonbuk	>8	4	≤1	CTX-M-14	-	CIP GEN NAL	-	-	A	P2
I-CF-5	I	Jeonbuk	>8	4	≤1	CTX-M-14	-	CIP GEN NAL	-	-	A	P1
I-CF-M-7	I	Jeonbuk	>8	4	≤1	CTX-M-14	-	CIP GEN NAL	-	-	A	P2
I-CF-M-8	I	Jeonbuk	>8	2	≤1	CTX-M-14	-	CIP GEN NAL	-	-	A	P1
K-CF-M-1	K	Jeonbuk	>8	>32	>16	CMY-2	-	CHL CIP NAL TET	+	I1α, FIB, FⅡ	D	ND
K-CF-7	K	Jeonbuk	>8	32	16	CMY-2	-	CIP NAL	-	-	D	ND
M-CF-M-3	M	Jeonnam	>8	8	8	CTX-M-55	-	CIP NAL	-	-	B1	ND
N-CF-5	N	Chungbuk	>8	>32	16	CTX-M-55, CMY-2	-	CHL CIP NAL STR FIS	+	K, X4	D	P27
O-CF-4	O	Gyeongbuk	>8	4	2	CTX-M-1	-	NAL FIS TET	+	I1α	A	P10
P-CF-M-2	P	Gyeongbuk	>8	8	4	CTX-M-55	-	CIP NAL	+	FⅡ	A	P20
P-CF-9	P	Gyeongbuk	>8	2	≤1	CTX-M-14	-	CIP GEN NAL TET	-	-	A	P4
T-CF-M-3	T	Chungbuk	>8	4	2	CTX-M-1	-	CHL CIP NAL FIS TET	+	I1α	B1	P21
V-CF-M-2	V	Jeonnam	>8	4	8	CTX-M-55	-	CHL CIP NAL STR FIS TET	-		B1	P11
W-CF-M-5	W	Jeonbuk	>8	4	8	CTX-M-55	-	CIP GEN NAL TET	+	I1-α, FIB, FⅡ	D	P24
W-CF-6	W	Jeonbuk	>8	8	8	CTX-M-55	-	CHL CIP NAL TET	+	FⅡ	B1	P14
W-CF-M-10	W	Jeonbuk	>8	32	>16	CTX-M-55, CMY-2	-	CHL CIP NAL STR FIS TET SXT	-		B1	P25
Y-CF-4	Y	Jeonbuk	>8	4	8	CTX-M-55	*qnrS1*	TET	+	FⅡ	B1	P5
Z-CF-M-1	Z	Chungnam	8	>32	8	CMY-2	-	CHL CIP NAL STR FIS TET	+	K	B1	P29
Z-CF-7	Z	Chungnam	>8	4	2	CTX-M-1	-	CIP NAL TET	-		B1	P28
AA-CF-M-2	AA	Chungnam	>8	4	4	CTX-M-55	-	CIP NAL	+	FIB, FⅡ	B1	P23
AB-CF-M-1	AB	Jeonbuk	>8	>32	16	CMY-2	-	CHL CIP NALSTR FIS TET	+	I1α	D	P33
AC-CF-2	AC	Jeonbuk	>8	32	>16	CTX-M-55	-	CHL CIP NAL FIS TET	-		B1	P13
AC-CF-M-3	AC	Jeonbuk	>8	4	4	CTX-M-55	-	CHL CIP NAL STR FIS SXT	+	FⅡ	A	P15
AC-CF-M-5	AC	Jeonbuk	>8	8	16	CTX-M-55	-	CHL CIP NAL FIS TET	-		B1	P13
AC-CF-10	AC	Jeonbuk	>8	2	≤1	CTX-M-14	-	CIP GEN NAL	-		A	P3
AD-CF-M-2	AD	Gyeonggi	8	8	≤1	CTX-M-14	-	CHL CIP GEN NAL FIS TET SXT	+	HI2, I1α	B1	P18
AD-CF-8	AD	Gyeonggi	8	16	≤1	CTX-M-14	-	CHL CIP GEN NAL STR FIS TET	+	HI2	B1	P16
AD-CF-M-10	AD	Gyeonggi	>8	4	8	CTX-M-55	-	CHL CIP NAL STR FIS TET SXT	-		A	P8
AE-CF-M-1	AE	Jeonnam	>8	>32	16	CMY-2	-	CHL CIP GEN NAL STR FIS TET SXT	+	I1α	B1	P30
AE-CF-4	AE	Jeonnam	>8	2	≤1	CTX-M-14	-	CIP GEN NAL	-		A	P3
AE-CF-M-9	AE	Jeonnam	>8	8	≤1	CTX-M-65	*qnrS2, aac(6′)-Ib-cr*	CHL CIP GEN NAL STR FIS TET SXT	-		B1	P12
AE-CF-M-10	AE	Jeonnam	>8	8	2	CTX-M-65	*aac(6′)-Ib-cr*	CHL CIP GEN NAL STR FIS TET SXT	-		B1	P12
AF-CF-M-1	AF	Jeonbuk	8	4	≤1	CTX-M-14	-	CHL CIP GEN NAL FIS TET SXT	+	HI2	B1	ND
AF-CF-M-4	AF	Jeonbuk	>8	4	≤1	CTX-M-14	-	CHL CIP GEN NAL FIS TET	+	HI2	B1	ND
AG-CF-M-4	AG	Jeonbuk	>8	8	8	CTX-M-55	-	CHL CIP NAL STR FIS TET SXT	+	FⅡ	A	P19
AG-CF-6	AG	Jeonbuk	>8	4	8	CTX-M-55	-	CHL CIP NAL STR FIS TET SXT	+	FⅡ	A	P19
AG-CF-M-9	AG	Jeonbuk	>8	8	8	CTX-M-55	-	CHL CIP NAL STR FIS TET SXT	+	FⅡ	A	P19
AG-CF-M-10	AG	Jeonbuk	>8	4	4	CTX-M-55	-	CHL STR FIS TET	+	I1α, FⅡ	B1	P17

AMC, Amoxicillin/clavulanic acid; CAZ, Ceftazidime; CHL, chloramphenicol; CIP, ciprofloxacin; COL, colistin; FOX, cefoxitin; GEN, gentamicin; NAL, nalidixic acid; STR, streptomycin; FIS, sulfisoxazole; TET, tetracycline; SXT, Trimethoprim/sulfamethoxazole; XNL, ceftiofur. PMQR, plasmid-mediated quinolone resistance. *Xbal* macrorestriction analysis yielded no DNA banding patterns in five *E. coli* isolates due to constant autodigestion of the genomic DNA, and thus, a cluster formed by this strain is excluded (ND, not determined). The underlined resistance markers were transferred to the recipient *E. coli J53* strain by conjugation.

**Table 5 microorganisms-08-01434-t005:** Plasmid replicon types and genetic environments of ESBL/pAmpC gene-carrying *E. coli* transconjugants.

Type of *bla* Gene	No. of Isolates	Self-Transfer	No. of Replicon Type										Genetic Environment			
			I1α	I1α+ FⅡ	I1α + FⅡ + FIB	FⅡ	FⅡ + FIB	HI2	HI2 + I1α	HI2 + FⅡ + Iγ	K	B/O +K + X4	Upstream		Downstream	
													IS*Ecp1*	IS*26*	IS*903*	*orf477*
CTX-M-1	4	3	2	-	1	-	-	-	-	-	-	-	3	-	-	3
CTX-M-14	12	5	-	-	-	-	-	3	1	1	-	-	-	-	5	-
CTX-M-55 ^a^	11	9	-	1	-	8	-	-	-	-	-	-	-	-	-	9
CTX-M-55 ^b^	7	4	-	1	1	1	1	-	-	-	-	-	4	-	-	4
CTX-M-65	2	0	-	-	-	-	-	-	-	-	-	-	-	-	-	-
CMY-2	8	6	3	1	1	-	-	-	-	-	1	-	6	-	-	-
CTX-M-55+CMY-2	2	1	-	-	-	-	-	-	-	-	-	1	1	-	-	-
Total	46	28	5	3	3	9	1	3	1	1	1	1	14	0	5	16

^a^*bla*_CTX-M-55_-*orf477*; ^b^*ISEcp1-bla*_CTX-M-55_-*orf477*.

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
