# Peer review of "Resistance Profiling and Molecular Characterization of Extended-Spectrum/Plasmid-Mediated AmpC β-Lactamase-Producing Escherichia coli Isolated from Healthy Broiler Chickens in South Korea"

_microorganisms, 2020, doi:10.3390/microorganisms8091434_

Round 1
Reviewer 1 Report
The article is devoted to a relevant topic, well written, adequate methods are used in the work. However, MLST results are clearly lacking. There are no serious comments on the work. From my point of view, some changes are advisable in the discussion section. It is possible to shorten the detailed comparison of the frequency of resistance and the distribution of the mechanisms of resistance in different regions. But it would be very important to compare the characteristics of the isolates included in the present study with the isolates that cause infections in people in South Korea (One Health approach). Absence of MLST should be mentioned as a limitation of the study.
Author Response
We are grateful for the constructive comments and suggestions on our manuscript. We believe that your comments and suggestions are essential to improve the quality of our manuscript and therefore, we have modified it accordingly. Our responses to the raised questions/comments are summarized below.
Response to Reviewer 1 comments
Point 1. The article is devoted to a relevant topic, well written, adequate methods are used in the work. However, MLST results are clearly lacking. There are no serious comments on the work. From my point of view, some changes are advisable in the discussion section. It is possible to shorten the detailed comparison of the frequency of resistance and the distribution of the mechanisms of resistance in different regions. But it would be very important to compare the characteristics of the isolates included in the present study with the isolates that cause infections in people in South Korea (One Health approach). The absence of MLST should be mentioned as a limitation of the study.
Response 1. Thank you for the constructive comments and suggestions on our manuscript. Based on your comments, the comparison of the frequency of resistance and the distribution of the mechanisms of resistance in different regions is revised (Lines 213-215, 223-224, 230-231, 243-244, and 262). Based on the importance of this study on “One-Health Approach” we have tried to compare the characters of the isolates in this study with those reported previously in humans in Korea and other countries, in each section (Lines 238, 244, 250, 262, 270, and 276). Indeed, comparisons were made in a sense that whether E. coli isolates harboring the same β-lactam resistance genes and other resistance determinants were associated with human infections in Korea and other countries. Unfortunately, taking the objective of this study and the information we have into, we are unable to make a detailed comparison of the genetic profiles of our isolates with that from humans. Further, the absence of MLST data is mentioned in the conclusion section (line 315), as a limitation.
Reviewer 2 Report
Antibiotic resistance of Escherichia coli strains isolated from chickens originating from 8 farms in Korea was analyzed and compared to plasmid profile. This is a classical microbiological work which adds some new data to our knowledge on microbial antibiotic resistance in chicken. The experiments were simple, but performed according to standards. It is not a breakthrough, but may contribute to the picture of contamination of chickens with antibiotic resistant bacteria.
I have only a few minor comments and suggestions:
- Line 60: Please, describe in more details farms which were sources of chickens. Were there large industrial farms or small backyard farms, or anything in between? This is importat to describe conditions of animal breeding as they might influence gut microbioms.
- Table 3 - Please, add a caption, common to columns 4-9, which should indicate what do the presented numbers mean.
- Lines 215-216: The sentence "However, it was lower......" does not make sense. What was lower? Previous sentence concerns isolates, rates, antibiotics and countries (all plular), so what does "it" mean?. If you mean rate of resistance, then it cannot be lower than "reports" - it might be lower than "those described in reports". Please, clarify.
- Lines 235-237: there are the same problems as that described in p.3.
- Line 273: Replace "is" with "are".
- Line 332: Replace "E. Coli" with "E. coli".
- In Discussion, some schemes would facilitate understanding of analyses, particularly comparisons between results obtained and described in various countries.
Author Response
We are grateful for the constructive comments and suggestions on our manuscript. We believe that your comments and suggestions are essential to improve the quality of our manuscript and therefore, we have modified it accordingly. Our responses to the raised questions/comments are summarized below.
Point 1. Line 60: Please, describe in more detail farms which were sources of chickens. Were there large industrial farms or small backyard farms, or anything in between? This is important to describe conditions of animal breeding as they might influence gut microbiomes.
Response 1. Chickens were originated from commercial broiler chicken farms (line 58-59)
Point 2. Table 3 - Please, add a caption, common to columns 4-9, which should indicate what do the presented numbers mean.
Response 2. A caption is included at the end of the table “The numbers in column 4-9 represent the number of isolates that carried blaCTX-M and blaCMY-2”.(Line 162)
Point 3 Lines 215-216: The sentence "However, it was lower......" does not make sense. What was lower? The previous sentence concerns isolate, rates, antibiotics, and countries (all plular), so what does "it" mean? If you mean rate of resistance, then it cannot be lower than "reports" - it might be lower than "those described in reports". Please, clarify.
Response 3. ”it" refers to the rate of resistance. Based on your suggestion, this section is revised as “However, it was lower than those described in recent reports in Asia and Africa’ (Line 213)
Point 4. Lines 235-237: there are the same problems as that described in p.3.
Response 4. Corrected as follows “However, it was higher than those reported in several Asian countries’(Line 223-224)
Point 5. Line 273: Replace "is" with "are".
Response 5. Corrected accordingly (Line 259)
Point 6. Line 332: Replace "E. Coli" with "E. coli".
Response 6. Corrected accordingly (Line 309)
Point 7. In Discussion, some schemes would facilitate understanding of analyses, particularly comparisons between results obtained and described in various countries.
Response 7. Based on your comments, we revised the discussion on the comparison of the frequency of resistance and the distribution of the mechanisms of resistance (Lines 213-215, 223-224, 230-231, 243-244, and 262).